# Laboratory Risk Assessment of Three Entomopathogenic Fungi Used for Pest Control toward Social Bee Pollinators

**DOI:** 10.3390/microorganisms10091800

**Published:** 2022-09-07

**Authors:** Mariana O. G. Leite, Denise A. Alves, Antoine Lecocq, José Bruno Malaquias, Italo Delalibera, Annette B. Jensen

**Affiliations:** 1Department of Entomology and Acarology, “Luiz de Queiroz” College of Agriculture, University of São Paulo, Avenida Pádua Dias 11, Piracicaba 13418-900, SP, Brazil; 2Department of Plant and Environmental Science, University of Copenhagen, Thorvaldsensvej 40, 1871 Frederiksberg, Denmark; 3Department of Biostatistics, Institute of Biosciences, São Paulo State University, Rua Prof. Dr. Antônio Celso Wagner Zanin 250, Botucatu 18618-689, SP, Brazil

**Keywords:** biopesticides, toxicology, stingless bees, honey bees, bumble bees

## Abstract

The use of fungal-based biopesticides to reduce pest damage and protect crop quality is often considered a low-risk control strategy. Nevertheless, risk assessment of mycopesticides is still needed since pests and beneficial insects, such as pollinators, co-exist in the same agroecosystem where mass use of this strategy occurs. In this context, we evaluated the effect of five concentrations of three commercial entomopathogenic fungi, *Beauveria bassiana*, *Metarhizium anisopliae*, and *Cordyceps fumosorosea,* by direct contact and ingestion, on the tropical stingless bees *Scaptotrigona depilis* and *Tetragonisca angustula*, temperate bee species, the honey bee *Apis mellifera*, and the bumble bee *Bombus terrestris,* at the individual level. Furthermore, we studied the potential of two infection routes, either by direct contact or ingestion. In general, all three fungi caused considerable mortalities in the four bee species, which differed in their response to the different fungal species. *Scaptotrigona depilis* and *B. terrestris* were more susceptible to *B. bassiana* than the other fungi when exposed topically, and *B. terrestris* and *A. mellifera* were more susceptible to *M. anisopliae* when exposed orally. Interestingly, increased positive concentration responses were not observed for all fungal species and application methods. For example, *B. terrestris* mortalities were similar at the lowest and highest fungal concentrations for both exposure methods. This study demonstrates that under laboratory conditions, the three fungal species can potentially reduce the survival of social bees at the individual level. However, further colony and field studies are needed to elucidate the susceptibility of these fungi towards social bees to fully assess the ecological risks.

## 1. Introduction

In recent decades, the use of natural biocides has increased as an eco-friendly alternative to chemical pest control in agricultural production [1,2]. Most commercialized products are based on hypocrealean entomopathogenic fungi (EF) and play a key role in integrated pest management programs (IPM) and organic farming [3]. Currently, EF comprises a significant slice of important markets in Brazil, the USA, and Europe [2,4,5]. The inundative application is the most used strategy [6,7], with a massive release of fungal conidia on crops, such as coffee [8], citrus [9], blueberries [10], and tomato [11]. These crops are known to use high levels of EF for pest control, but at the same time, they rely on wild and managed pollinators to improve yield and/or quality [12].

It is essential to understand the possible interactions between the bioagents and pollinators, which so far there is a substantial knowledge gap [13,14]. Under favorable environmental conditions, hypocrealean EFs are considered generalists as they can infect and multiply on a broad spectrum of insect hosts [15]. The primary infection route is through the cuticle when insects are directly exposed to fungal conidia [16]. However, infections can also occur orally or through other body openings [17]. Since high amounts of EF are applied in crop fields, non-target insects can be directly exposed to fungal spores during the application or indirectly exposed when in contact with contaminated leaves, soil, or during foraging activity for nectar and pollen collection [18,19,20].

Among the primary crop pollinators, bees have a prominent role. Of more than 20,000 described species worldwide, a small fraction of them is managed for crop pollination, such as the Western honey bee, some bumble bees, and stingless bee species [21,22]. Given the pivotal role of these social bees in agroecosystems [23] and the increasing use of EF for pest control, risk assessment of EF’s impact on bees is crucial for ensuring more sustainable agricultural practices. In order to minimize the potential environmental risks associated with EF, the Food and Agriculture Organization (FAO) created the International Standards for Phytosanitary Measures No. 3, including the need to carry out risk assessment studies for non-target organisms [24]. Yet, most evaluations of the effect of biopesticides focus on the honey bee *Apis mellifera* [25,26,27,28,29,30], while bumble bees [11,31], stingless bees [32,33], and solitary bees [34] have received much less attention [35,36].

In such a complex model as the agroecosystem, where EF interact with the target organisms but also with the pollinators, it is critical to understand the responses of multiple bee species to the same strategy of biocontrol. Based on their capacity to infect a wide range of insect hosts by different routes, we hypothesize that EF could potentially harm social bees. The assay was performed at the individual level as it is the most standardized process for biopesticides risk assessments [35,36] and due to the foragers being directly exposed to EF when foraging. More specifically, at laboratory conditions, our study aims are (a) to evaluate the individual direct effect of three of the most commercialized fungal-based biopesticides, *Beauveria bassiana*, *Metarhizium anisopliae,* and *Cordyceps fumosorosea,* on the survival of four social bees, native from tropical (the stingless bees *Scaptotrigona depilis* and *Tetragonisca angustula*) and temperate regions (the honey bee *A. mellifera*, and the bumble bee *B. terrestris*), (b) at a range of five concentrations, as recommended doses of EF application in crop fields [9,10] and (c) by topical and oral exposure.

## 2. Materials and Methods

### 2.1. Stingless Bees

The study was carried out between August and December 2019, using *S. depilis* and *T. angustula* colonies. The colonies (five colonies for each species) were maintained in free-foraging wooden nest boxes in an outdoor meliponary shelter at the Department of Entomology and Acarology of the “Luiz de Queiroz” College of Agriculture (ESALQ) at the University of São Paulo (USP), Piracicaba, Brazil. Before the bioassay began, each colony was checked visually for the absence of diseases or pests.

For *S. depilis*, we sampled brood combs with mature pupae and placed them in a wooden box in an incubator (28 ± 1 °C, 70 ± 5 % RH, 0:24 L:D), allowing us to collect all newly emerged workers and controlling the age [37]. Daily, the newly emerged workers were moved to a wooden box with syrup (1:1 *w/v*, organic sugar: water) ad libitum and maintained in the same conditions from 12 to 17 days, when they were fully melanized. For the bioassay, five 12–17-day old workers were then carefully transferred with a soft tweezer to a plastic cage (2 cm high, 15 cm diameter) lined with a paper filter, for a total of 36 plastic cages (3 EF × 2 methods of application × 6 concentrations) per colony.

For *T. angustula*, we collected pollen foragers returning to their colonies between 7:00 and 10:00 h. Subsequently, foragers were chilled for a few seconds at 5 °C to immobilize them and transferred to a wooden box maintained in the same conditions mentioned for *S. depilis*. For the bioassays, eight workers were transferred with a soft tweezer to a plastic cage (2 cm height × 15 cm diameter) lined with filter paper, for a total of 36 plastic cages (3 EF × 2 methods of application × 6 concentrations) per colony. Since the *T. angustula* broods are very delicate, we used foragers instead.

### 2.2. Honey Bees and Bumble Bees

The experiments with *A. mellifera* and *B**. terrestris* were carried out from May to August 2020 in the Department of Plant and Environmental Science at the University of Copenhagen (KU), Copenhagen, Denmark. For *A. mellifera,* combs containing mature worker pupae were collected from five hives of the experimental apiary on the campus and maintained in an incubator at 30 ± 1 °C, 70 ± 5% RH, and 0:24 L:D, until the emergence of bees. The newly emerged workers were moved with a soft tweezer to a plastic cage (12.5 cm height × 10 cm diameter) lined with filter paper and supplied with sugar solution (1:1 *w/v*, organic sugar: water) ad libitum. For the bioassays, ten 4-day-old workers were transferred to a new plastic cage (12.5 cm height × 10 cm diameter) lined with filter paper, repeated for a total of 36 plastic cages (3 EF × 2 methods of application × 6 concentrations) per colony.

For *B. terrestris*, five colonies were purchased from EWH Bioproduction, Tappernøje, Denmark, and kept in standard laboratory conditions (22 ± 2 °C and 65% RH). They were weekly fed with irradiated sterilized honey bee pollen and sugar solution (1:1 *w/v*). Each nest was opened inside a dark room under red light to prevent bees from flying off. For the bioassays, five workers were caught with 25 cm long tweezers and put into a plastic cage (12.5 cm height × 10 cm diameter) lined with filter paper, repeated for a total of 36 plastic cages (3 EF × 2 methods of application × 6 concentrations) per colony.

### 2.3. Fungal Material

The fungi *M. anisopliae* E9 (Ma), *B. bassiana* PL63 (Bb), and *C. fumosorosea* 1296 (Cf), maintained at –80 °C, were provided by the Collection of Entomopathogenic Microorganisms of the Laboratory of Pathology and Microbial Control of Insects, in the Department of Entomology and Acarology, ESALQ-USP. Conidia were produced on Potato Dextrose Agar (PDA, Difco®, Piracicaba, Brazil). They were harvested from each fungus by scraping the surface of the agar plates with a glass rod and rinsing it in glass tubes with 10 mL sterile distilled water containing 0.05% Tween 80. The glass tubes were sealed and vortexed for 1 min to produce a homogenous conidial suspension. A serial dilution (4×) of the conidial suspension was prepared to determine the concentration. From the lowest suspension, 180 µL was pipetted on a Neubauer hemocytometer and adjusted to 0 (control, C0), 5 × 10^5^ conidia mL^−1^ (C1), 1 × 10^6^ conidia mL^−1^ (C2), 5 × 10^6^ conidia mL^−1^ (C3), 1 × 10^7^ conidia mL^−1^ (C4), 5 × 10^7^ conidia mL^−1^ (C5) in sterile distilled water. All conidial suspensions were maintained at 4 °C for no longer than 24 h before use.

### 2.4. Fungal Exposure Bioassay

To test the susceptibility of four social bees to *B. bassiana*, *M. anisopliae,* and *C. fumosorosea,* we used five concentrations of each fungus by both topical and oral exposure. Both exposure methods have been reported as methods for bioproduct risk assessments [35].

For topical exposure, 1 µL of the conidia suspension was applied to the pronotum area of each worker, which was held for 10 s to allow the drop to spread. Due to the differences in body sizes across species, the 1 µL drop represented a different dose/area for each bee species, but each worker got the same dose. Workers were then held in a plastic cage (five *S. depilis* workers/cage; eight *T. angustula* workers/cage; ten *A. mellifera* workers/cage; five *B. terrestris* workers/cage) at 22 ± 2 °C and 65% RH and provided with sugar solution (1:1 *w/v*) ad libitum.

For oral exposure, stingless bee workers were individualized in 3 cm glass Petri dishes containing an open reservoir filled with 200 µL of the fungi solution mixed with sugar (1:1 *w/v*) which assured ad libitum consumption for 24 h. After 24 hours, workers of each stingless bee species were gently moved with a soft tweezer to a 15 cm plastic cage lined with filter paper and containing sugar solution (1:1 *w/v*) ad libitum. Each cage had eight *T. angustula* workers or five *S. depilis* workers.

For honey bees and bumble bees, workers were kept in cages with a plastic tube filled with 1 mL of the conidia suspension mixed with sugar solution (1:1 *w/v*)—workers had free access to the reservoir through a small hole of 0.5 mm drilled in the lid, as described by [38]. After 24 h, the plastic tube reservoir was substituted with sugar solution (1:1 *w/v*) ad libitum. Each cage had ten *A. mellifera* workers and five *B. terrestris.*

For the topical application, the fungal dose was kept controlled at 1 µL/worker, whereas for the oral exposure the precise dose could not be controlled since the fungus-sugar mix was offered freely to the bees. In this case, it was assumed that the fungal dose ingested by each worker varied according to their body size. All assays were carried out for 7 days, and the mortality rate was evaluated daily. The dead bodies were surface sterilized with 1× sodium hypochlorite, 1× 70% ethanol, and 3× distilled water and put in a humid chamber, individually, in a 60 × 15 mm plastic plate lined with a moistened cotton wool, to verify fungal conidiogenesis [39]. The dead bees were incubated at 25 ± 2 °C, 65% RH, 0:24 L:D, and mycosis was evaluated 2 to 7 days after fungal exposure. The fungal sporulation and consequently mortality by the fungus was confirmed by the presence of white, green, or light purple colored conidia for *B. bassiana*, *M. anisopliae,* and *C. fumosorosea*, respectively. We made five replicates for all the fungi treatments, and the number of replicates was the same for both methods of application and the four bee species.

### 2.5. Statistical Analysis

The effects of the entomopathogenic fungi on workers’ survival were assessed using Weibull regression survival model. The multiple comparisons of survival curves and the pairwise comparisons between group levels with corrections for multiple testing were performed with R packages *survminer* [40] and *survival* [41]. Corrected mortality was assessed using a Bayesian model estimation [42]. The comparisons of mortality curves were performed with *Multicomp* package [43]. Data of EF concentration were transformed by log10(×) and then fitted to a generalized linear model (GLM) with binomial distribution considering overdispersion and a logit link function. Fixed effects attributed to fungal isolates and concentrations in the model were assessed for significance with *F*-tests. In all bioassays, mortality was recorded and monitored daily for seven days after the fungal application. Mortality due to the fungal treatment was confirmed and expressed as mycosis (fungal outgrowth) level. Data of concentration mycosis correlation were transformed by log10(×) and then fitted to a generalized linear model (GLM) with binomial distribution considering overdispersion and a logit link function. The comparisons of mycosis curves were performed with R *Multicomp* package [43]. Fixed effects attributed to fungal isolates and concentrations in the model were assessed for significance with *F*-tests. All models chosen here to fit these datasets were carefully selected based on their goodness-of-fit, using residual plots and half normal plots [44].

## 3. Results

### 3.1. Effect of EF on the Survival of Bees

The survival effect from the interaction between EF, bee species, and method of exposure was significant (X^2^ = 18.01, df = 6, *p* = 0.0062). With regards to the different levels of susceptibility for each fungus among the bee species, we found that the fungus *B. bassiana* highly affected *S. depilis* survival when topically administrated (X^2^ = 23.291, df = 3, *p* < 0.0001). Yet, when *B. bassiana* was orally administrated, it decreased the survival of *S. depilis*, *T. angustula,* and *B. terrestris* (X^2^ = 9.959, df = 3, *p* = 0.0189). The fungus *C. fumosorosea* reduced both stingless bees and *A. mellifera* lifespan when topically applied (X^2^ = 16.672, df = 3, *p* = 0.0008), while *B. terrestris* was highly affected when *C. fumosorosea* was orally administrated (X^2^ = 17.949, df = 3, *p* = 0.0004). The EF *M. anisopliae* only significantly affected the *T. angustula* bees when topically applied (X^2^ = 18.732, df = 3, *p* = 0.0003), but when orally administrated, it affected the survival of *S. depilis*, *T. angustula*, and *B. terrestris* (X^2^ = 12.889, df = 3, *p* = 0.001) (Table 1).

Bee survival was significantly reduced after the exposure to the three EF and both application methods, except for *B. terrestris* treated by topical exposure, where the fungus did not considerably reduce bumble bee survival compared to the untreated control (Figure 1 and Figure 2). 

For the stingless bee *S. depilis*, the workers had their survival significantly reduced when topically (*X²* = 32.4, df= 5, *p* < 0.0001) and orally (*X²* = 20.2, df = 5, *p* = 0.0001) exposed to *B. bassiana* by all the concentrations. Topically administrated *M. anisopliae* (*X²* = 20.0, df = 5, *p* = 0.006) reduced survival with concentrations C1 (*p* = 0.0065) and C5 (*p* = 0.0018) meanwhile *C. fumosorosea* (*X²* = 2.3, df = 5, *p* = 0.046) affected *S. depilis* survival with the highest C5 (*p* = 0.049) (Figure 1). Topically administrated *B. bassiana* was the most virulent (0.76 ± 0.43, *p* = 0.0002) to *S. depilis* workers (Appendix A). The orally administrated *M. anisopliae* (*X²* = 47.8, df = 5, *p* < 0.0001) and *C. fumosorosea* (*X²* = 43.6, df = 5, *p* < 0.0001) affected the survival from the concentration C2 (Ma; C2: *p* = 0.0001; C3: *p* = 0.0182; C4: *p* < 0.0001; C5: *p* < 0.0001) and (Cf; C2: *p* = 0.0094; C3: *p* = 0.0007; C4: *p* = 0.0006; C5, *p* < 0.0001; Figure 1). 

The *T. angustula* workers had their survival significantly reduced when topically (Bb: *X²* = 17.6, df = 5, *p* = 0.0002; Ma: *X²* = 20.5, df = 5, *p* = 0.0003; Cf: *X²* = 20.5, df = 5, *p* = 0.0006) and orally exposed to EF (Bb: *X²* = 34.0, df = 5, *p* < 0.001; Ma: *X²* = 74.2, df = 5, *p* < 0.001; Cf: *X²* = 62.4, df = 5, *p* < 0.001) (Figure 1 and Figure 2).

After topical exposure, the survival of *A. mellifera* workers was also affected by *B. bassiana* (*X²* = 27.46, df = 5, *p* < 0.0001), *M. anisopliae* (*X²* = 3.931, df = 5, *p* < 0.0001) and *C. fumosorosea* (*X²* = 0.532, df = 5, *p* < 0.0001) at the three highest concentrations (Figure 1). When the three EF were orally administrated, they reduced the workers’ survival at all the concentrations (Bb: *X²* = 32.1, df = 5, *p* < 0.0001; Ma: *X²* = 76.5, df = 5, *p* < 0.0001; Cf: *X²* = 56.8, df = 5, *p* < 0.0001) (Figure 2). By ingestion, *M. anisopliae* and *B. bassiana* were the most virulent EF for *A. mellifera* workers (Ma: 0.52 ± 0.50; Bb: 0.58 ± 0.49, *p* = 0.0054; Appendix A).

The survival curves of *B. terrrestris* workers topically treated with EF were similar to the controls (Bb: *X²* = 7.3, df = 5, *p* = 0.2; Ma: *X²* = 5, df = 5, *p* = 0.4; Cf: *X²* = 9.1, df = 5, *p* = 0.1; Figure 1). However, the survival was drastically reduced when workers were fed with *B. bassiana* (*X²* = 14.5, df = 5, *p* < 0.0001), *M. anisopliae* (*X²* = 57.1, df = 5, *p* < 0.0001) and *C. fumosorosea* (*X²* = 17.9, df = 5, *p* < 0.0001). When topically administrated, *B. bassiana* was the most virulent EF for *B. terrestris* workers (0.34 ± 0.47, *p* = 0.0311), although *M. anisopliae* was the most virulent when orally offered (0.92 ± 0.27, *p* = 0.0006; Appendix A).

### 3.2. Sporulation of Entomopathgenic Fungi on Dead Bees

The method of fungal application had a significant effect on the fungal sporulation capacity on dead *S. depilis* (*p* = 0.0042), *T. angustula* (*p* = 0.02), *A. mellifera* (*p* = 0.0028), and *B. terrestris* (*p* < 0.0001) workers. In general, there was fungus outgrowth on dead corpses (Figure 3). Although the proportion of sporulated bees varied considerably among the EF, all three EF showed lower outgrowth proportion by topical application than by oral infection, especially at lower concentrations.

The controls did not present sporulation. There was a significantly higher proportion of sporulation in *S. depilis* workers when orally infected with *B. bassiana*, *M. anisopliae,* and *C. fumosorosea* (*p* < 0.0001) and when topically treated with *M. anisopliae* (*p* = 0.0172) and *C. fumosorosea* (*p* = 0.0011) when compared to the controls. Yet, sporulation had a marginally significant effect when *B. bassiana* was topically applied to workers (*p* = 0.0504; Figure 3). The fungal outgrowth on *T. angustula* and *A. mellifera* workers was significant for the three EF by both exposure methods (*p* < 0.01). However, *T. angustula* topically exposed by EFs showed a lower variation between concentrations, and *A. mellifera*-exposure showed an increased positive concentration response. The fungal sporulation on *B. terrestris* indicated that *M. anisopliae* (*p* = 0.133) and *C. fumosorosea* (*p* = 0.566), when topically applied, did not have any significant fungal outgrowth on workers, but it did have when *B. bassiana* was used (*p* = 0.0116). When these three fungi were ingested, sporulation was significantly higher than in the controls (*p* < 0.0001).

## 4. Discussion

Our study assesses the impact of entomopathogenic fungi on four social bee species at the individual level, at different concentrations, and under exposure methods. The effects of the fungal agents depended on the dose administered and the bee species for both exposure methods. The results revealed that all three entomopathogenic fungi, *B. bassiana*, *M. anisopliae*, and *C. fumosorosea,* significantly decreased *T. angustula*, *S. depilis*, and *A. mellifera* survival at different concentrations by both infection methods and lowered *B. terrestris* survival after oral exposure.

The stingless bees tended to be more affected by the three fungi. Both stingless bee species had a 50% decrease in survival when topically exposed, depending on the concentration, from the fourth to fifth day post-application. The fungus *B. bassiana* was most virulent to *S. depilis* followed by *M. anisopliae* and *C.*
*fumosorosea*, whereas all three EF had a similar high dose response on *T. angustula* (Appendix A). In a similar study, which tested the same three EF species, but different isolates directly applied at a very high concentration (10^9^ conidia mL^−1^), *M. anisopliae* affected *T. angustula*, *Melipona beecheii,* and *S**. mexicana* workers’ mortality (94%, 53%, and 38.9%, respectively). On the other hand, *B. bassiana* and *Cordyceps fumosorosea* (previously known as *Isaria fumosorosea*) caused less than 30% mortality for all three species [32]. *Melipona scutellaris* has also shown to be somewhat susceptible to *B. bassiana,* with mortality over 56% when topically exposed to 1 × 10^5^ conidia mL^−1^ [33]. However, different isolates of *B. bassiana* and *M. anisopliae*, when directly applied in a high concentration (10^9^ conidia mL^−1^), did not cause significant mortality in *Meliponula ferruginea* [45].

For *A. mellifera* and *B. terrestris*, the three EF species affected the worker’s survival, depending on the route of infection. When orally exposed, all EF species caused a significant effect on *A. mellifera* and *B. terrestris* survival, while topical exposure did not significantly affect *B. terrestris*. The three EF reduced *A. mellifera* workers’ survival by 50% from the sixth day after application when topically exposed, and from the third to fourth day when orally exposed. When Africanized *A. mellifera* workers were directly sprayed or orally fed with other isolates of *B. bassiana* and *M. anisopliae* (10^9^ conidia mL^−1^), both reduced workers’ survival, with a faster response when bees were sprayed than orally fed [46]. Both EF also caused *A. mellifera* mortality above 50% after the fifth day by direct and oral exposure [28]. In this study, the EF *B. bassiana* and *M. anisopliae*, at 10^7^ conidia mL^−1^, caused more than 50% mortality. Topical exposure of several *B. bassiana* and *M. anisopliae* strains at 10^7^ conidia mL^−1^ on *A. mellifera* workers resulted in mortalities from 40% to 100% [47]. Significantly, deaths were seen in *B. terrestris* adults when topically exposed to 10^8^ CFU mL^−1^ of *B. bassiana-* and *C. fumososrosea*-based products [48] and to *B. terrestris*, *B. lucorum*, and *B. lapidarius* when exposed to 10^8^ conidia mL^−1^ of *M. anisopliae* [49]. The EF *B. bassiana* affected the lifespan of *B. terrestris* workers, decreasing by up to 4 days at 18 °C and by 13 days at 28 °C, by applying conidia over the whole worker body, without a specific concentration [50]. When newly emerged *B. terrestris* workers were exposed orally and topically to *B. bassiana*-based products, both routes caused mortality. However, contrary to our results, the topical method killed 92% of the workers meanwhile the oral killed less than 30% [51].

Regarding risk assessment of fungal-based products on social bees, we still face a lack of substantial knowledge about its lethal (and sub-lethal) effects. Since most of the published studies only focus on the Western honey bee, the potential effects of EF on non-*Apis* managed and wild species, which are an important and untapped group of crop pollinators, remain largely unexplored [36]. In this sense, understanding the potential effects that the same EF might cause on different bee species is essential to properly develop the regulation and use of biopesticides for pest management. Here, we show that the bee species responded differently to the EF and it might be due to their morphological traits, which can interfere with the effectiveness of a fungal infection. Our data show that the stingless bees were more affected by the entomopathogenic fungi when in direct contact than the honey bees and bumble bees. Due to their small body size, the same drop size of the fungal suspension resulted in a higher dose per total body area of the stingless bees than for the honey bees and bumble bees, even though the number of conidia/drops was the same. Additionally, the two studied stingless bee species are less hairy than *A. mellifera* and *B. terrestris*. During the infection process, EF conidia interact with their environment by electrostatic properties [52]. Conidial surfaces have a net negative charge that attracts them toward positively charged surfaces [53]. Since the bees have branched hairs [54] that have electrostatic forces [55], the conidia could be less likely to adhere to the cuticle, being attached to the hair and thus, not able to get in contact with the cuticle directly. Therefore, less hairy bees, like many stingless bee species, could be more susceptible to contact with fungal conidia and consequently suffer more from the infectious process when topically exposed.

Another critical point is the inter- and intra-specific variation in fungal virulence. Different isolates of EF species were used in the aforementioned studies, so the variable results might reflect different virulence of the fungal strains used, as it is known that virulence traits might vary within a single fungal species [56]. The mechanisms that led to different outcomes of EF virulence on social bee species and routes of infection were not examined in this study, but some possible speculations are suggested hereafter.

Regarding the EF’s virulence and dose used, the infection route showed to play a significant role in workers’ mortality. The cuticle represents the first point of contact and barrier between the fungus and the insect; however, it is known that the fungi can infect through other paths [17,57,58]. Indeed, the EF *M. anisopliae* caused higher mortality when orally offered, as seen in the study by [28] on Africanized honey bees. Furthermore, the EF *M. anisopliae* produces specific mucilage and adhesive proteins, increasing the facility to penetrate any part of the workers’ body [59,60], including the buccal parts. For example, the buccal cavity is a known site for *M. anisopliae* conidia to adhere, germinate on, and penetrate the sheep blowfly *Lucilia cuprina* [61], the pine weevil *Hylobius pales* [62], and the desert locusts *Schistocerca gregaria* [63]. Studies examining the adhesion of *B. bassiana* to surface substrata showed direct binding of conidia to hydrophobic surfaces [64], like most insect cuticles, which present a hydrophobic barrier rich in lipids [53,57]. Additionally, *B. bassiana* produces secondary metabolites acting as immunosuppressants, facilitating contact infection, such as beauvericin, bassianolide, oosporein, tenellin, bassiantin, and beauverolides [16].

Our results also showed that the different exposure methods affected the EF mortality differently in social bees, especially *B. terrestris*. When *B. bassiana* and *M. anisopliae* were orally offered to Africanized honey bees, with a dose of 10^8^ conidia mL^−1^, it caused more significant mortalities (90% and 84%, respectively), compared to when topically applied (84% and 26%) [65]. While the conidia have to activate all the germination and infection pathways through cuticular layers by topical application, oral exposure may take a shortcut for the infection. When bees ingest the fungal suspension, the conidia get in direct contact with the mouthparts, which are softer and with multiple intersegmental parts more susceptible to fungal entrance [17,66]. The higher mortality for oral exposure in our bioassays might also be due to bee body size, especially for *B. terrestris*. As *B. terrestris* are our largest study species (19–22 mm length), followed by *A. mellifera* (12–15 mm), *S. depilis* (6 mm), and *T. angustula* (4–5 mm), they were probably capable of consuming a higher volume of the fungal solution compared to honey bees and stingless bees, and thus, a higher amount of conidia over 24 h. Moreover, social bees display prophylactic behaviors against pathogens, such as allogrooming, whereby co-workers clean each other [67]. This behavior could have caused the dispersion of the fungi among worker bodies or even ingestion while they were cleaning each other [68], increasing mortality rates. An important point to be highlighted is that oral infection is commonly used to define mycosis through ingestion, but with no definition of whether this infection process occurred in the mouthparts or the intestinal tract [62].

Interestingly, we visually detected a cue of the entomopathogenic fungal infection on the workers before any external development. Entomopathogenic fungi infect and multiply within the insect hosts as hyphae, and after the host dies, the fungus becomes visible by hyphal growth and subsequent sporulation externally [39]. We observed that when dead bees were kept at room temperature for hours or days before putting them in a humid chamber, some of them developed a change in eye color, presumably due to the fungal growth. Subsequent sporulation, initially throughout the eye, was confirmed after incubation in a humid chamber. This happened mainly for *B. bassiana*, characterized by white eyes (Figure 4), but it was also observed in workers infected by *M. anisopliae,* and *C. fumosorosea*. Whether this symptom could be turned into a possible visual cue for infected bees in the crop field stays open for further studies.

Pesticide risk assessments are complex and even more complicated in social insect species, such as bees, because the main goal is not to evaluate features of a single individual but of the colony as a whole [69]. In our experiments, control workers showed some mortality, probably because of the absence of social interactions [70]. Moreover, in laboratory conditions, the insects are maintained in a non-natural environment that causes stress and favors the development of the fungi. Thus, even though the social bee species tested in this study showed a significantly reduced survival, the laboratory assay does not represent the reality in the field [71,72]. Honeybee colonies exposed to *Beauveria* sp. and *Metharhizium* sp. to control varroa mites were not affected negatively but instead increased the numbers of adult bees and brood production [73,74]. The infection process might be prevented by the social immune response of the bees [70].

Different from chemical pesticides, entomopathogenic fungi are naturally occurring generalist pathogens widespread in the soil, plant surface, and as endophytes [75]. Thus, they co-exist with social bees in natural settings and not only when applied as biopesticides. Social colonies are composed of close relatives living at high densities with frequent contact, making them especially susceptible to spreading diseases. In addition to these features, social bees are highly resistant to generalist pathogens mainly due to several defense mechanisms at the colony level [76]. These mechanisms can include behavioral, genetic, physiological, spatial, or morphological defenses [76,77] as well as the symbiotic association with microorganisms that protect against microbial pathogens [78]. Hence, more realistic assays, including the whole colony and its symbiotic elements, are needed to evaluate the safety of entomopathogenic fungi-based biopesticides towards non-target insect species.

Standardized protocols exist for honey bees (recognized by the Organization for Economic Cooperation and Development (e.g., [79,80]), stingless bees [81,82], and bumble bees [83,84] for toxicological assessments with chemical pesticides. However, for fungus-based biopesticides, there is still a lack of such protocols, even for the requirement of new product registration tests [85,86].

On the other side, entomopathogenic fungi biopesticides are a reliable alternative to chemicals. In some cases, they are one of the few alternatives [5], so risk assessment tests should be evaluated carefully. Risk assessment for social bees should also consider the challenges in the field scenario, considering the behavioral traits of bee species, the target crop, the time, and the method of each biopesticide application. One example is the use of *B. bassiana* in coffee crops in Brazil. Bees are expected to visit coffee plants during the flowering season and to rarely visit them outside it [87]. *Beauveria bassiana* is often applied mainly after the flowering season, but some applications can be made before this season (a coffee farmer, personal communication). Thus, it is likely that the pollinators will not be affected by the fungus application. Still, careful evaluation could help decide the best timing for biopesticide application, considering the insect pests and the pollinators.

## 5. Conclusions

In conclusion, this study demonstrates that the recommended concentrations of *B. bassiana, M. anisopliae*, and *C. fumosorosea* and, in some cases, even lower concentrations can potentially reduce individuals’ survival of social bees in laboratory conditions. Even though laboratory studies are a valuable tool for first-tier risk assessment, allowing an accurate evaluation of colony fitness parameters using controlled concentrations under standardized conditions [88], colony and field risk assessments are further needed.

## Figures and Tables

**Figure 1 microorganisms-10-01800-f001:**
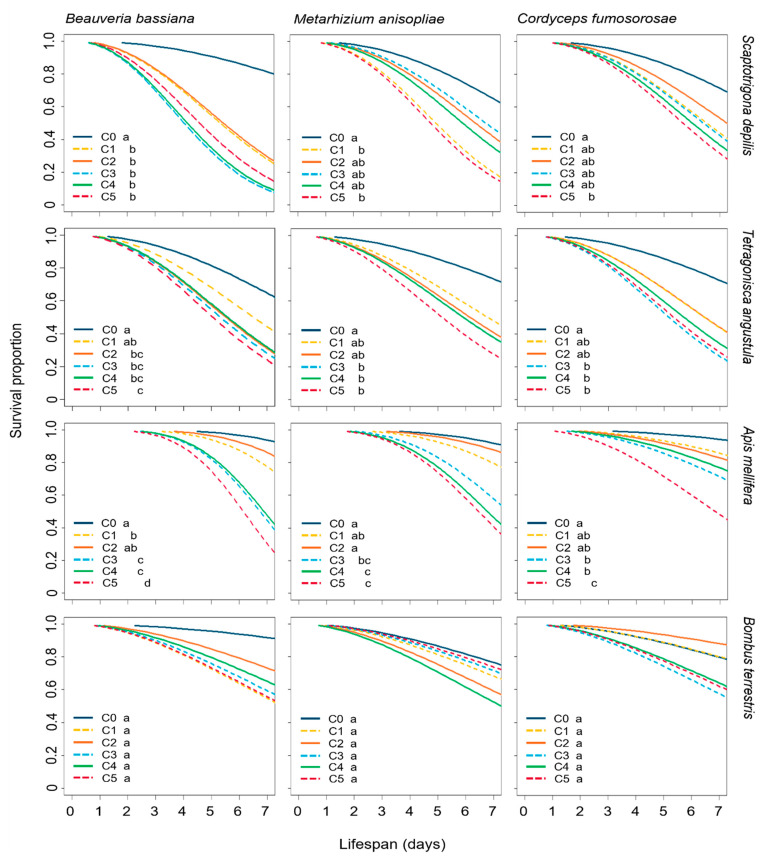
Survival proportion of workers (*Scaptotrigona depilis* (*n* = 150), *Tetragonisca angustula* (*n* = 240), *Apis mellifera* (*n* = 300) and *Bombus terrestris* (*n* = 150)) to topical application of three entomopathogenic fungi, *Beauveria bassiana*, *Metarhizium anisopliae* and *Cordyceps fumosorosea*. Concentrations: control (C0), 5 × 10^5^ conidia mL^−1^ (C1), 1 × 10^6^ conidia mL^−1^ (C2), 5 × 10^6^ conidia mL^−1^ (C3), 1 × 10^7^ conidia mL^−1^ (C4), 5 × 10^7^ conidia/mL^−1^ (C5). Concentrations with different letters are significantly different (*p* < 0.05).

**Figure 2 microorganisms-10-01800-f002:**
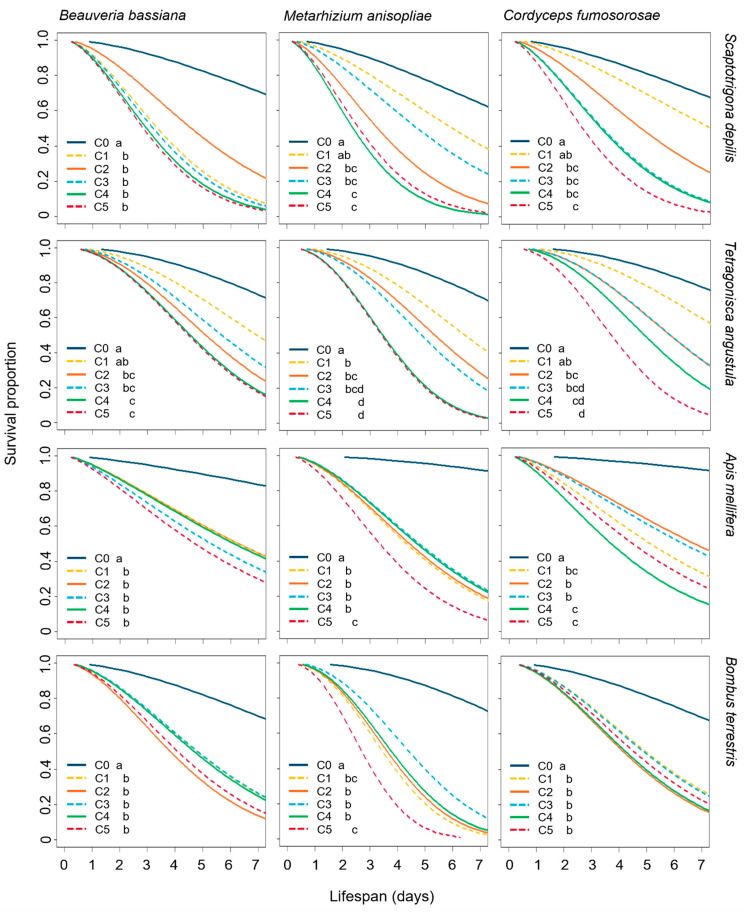
Survival proportion of *Scaptotrigona depilis* (*n* = 150 workers), *Tetragonisca angustula* (*n* = 240 workers), *Apis mellifera* (*n* = 300 workers) and *Bombus terrestris* (*n* = 150 workers) to oral exposure of three entomopathogenic fungi, *Beauveria bassiana*, *Metarhizium anisopliae* and *Cordyceps fumosorosea*. Concentrations: control (C0), 5 × 10^5^ conidia mL^−1^ (C1), 1 × 10^6^ conidia mL^−1^ (C2), 5 × 10^6^ conidia mL^−1^ (C3), 1 × 10^7^ conidia mL^−1^ (C4), 5 × 10^7^ conidia/mL^−1^ (C5). Concentrations with different letters are significantly different (*p* < 0.05).

**Figure 3 microorganisms-10-01800-f003:**
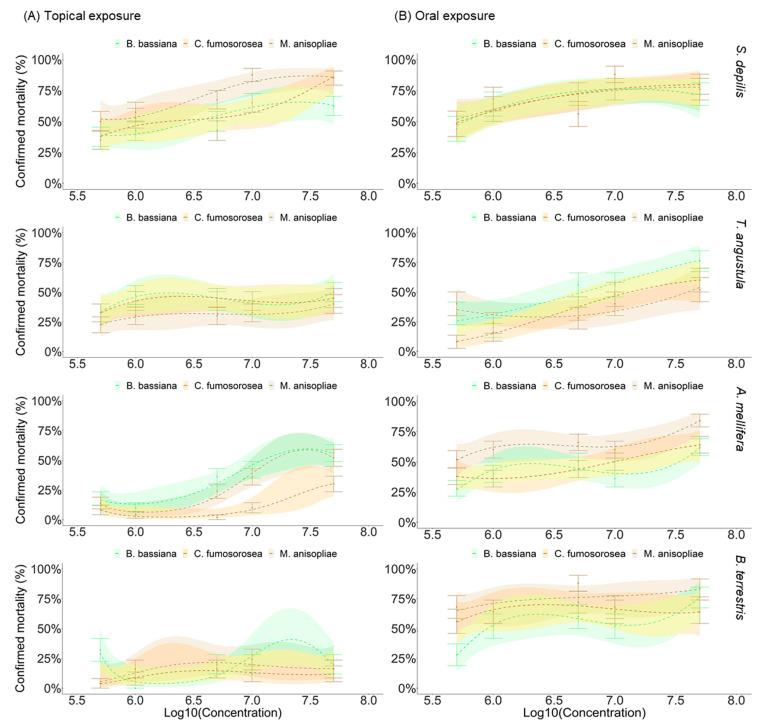
Mean (±SE) percentage of fungal outgrowth curve in dead *Scaptotrigona depilis*, *Tetragonisca angustula*, *Apis mellifera,* and *Bombus terrestris* workers topically (**A**) and orally (**B**) exposed to the fungi *Beauveria bassiana*, *Metarhizium anisopliae,* and *Cordyceps fumosorosea*. The concentrations (conidia mL^−1^) were transformed by regression Log10: 4.0 = 5 × 10^5^, 4.5 = 1 × 10^6^, 5.0 = 5 × 10^6^, 5.5 = 1 × 10^7^, 6.0 = 5 × 10^7^. Fitted to generalized linear model (GLM) with binomial distribution.

**Figure 4 microorganisms-10-01800-f004:**
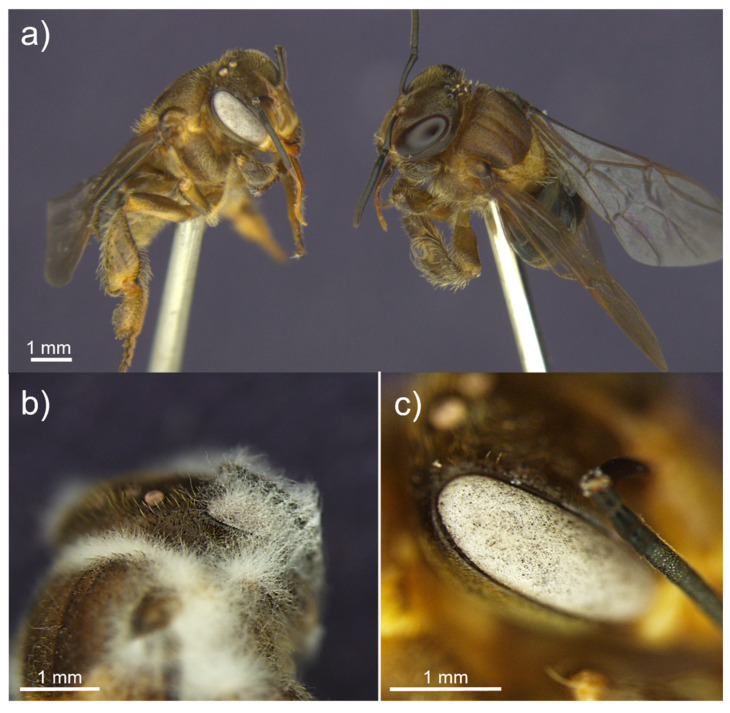
Comparison of the eye color change of a *B. bassiana*-infected (**left**) and non-infected (**right**) *Scaptotrigona depilis* (**a**). Detail of a *S. depilis* dead body with *B. bassiana* outgrowth (**b**). Detail of the *B. bassiana* growth inside *S. depilis* eye (**c**).

**Table 1 microorganisms-10-01800-t001:** Susceptibility of Scaptotrigona depilis, Tetragonisca angustula, Apis mellifera, and Bombus terrestris workers to the entomopathogenic fungi Beauveria bassiana, Cordyceps fumosorosea, and Metarhizium anisopliae for each exposure method (topical or oral). Mean mortality fitted to a generalized linear model (GLM) with binomial distribution, Tukey test.

Method	Fungi	Bee Species				*p*-Value
		*S. depilis*	*T. angustula*	*A. mellifera*	*B. terrestris*	
topical	*B. bassiana*	0.76 ^a^	0.48 ^b^	0.36 ^b^	0.34 ^b^	<0.0001
	*C. fumosorosea*	0.48 ^a^	0.48 ^a^	0.30 ^ab^	0.16 ^b^	0.0008
	*M. anisopliae*	0.38 ^b^	0.50 ^a^	0.22 ^bc^	0.14 ^c^	0.0003
oral	*B. bassiana*	0.80 ^a^	0.58 ^ab^	0.50 ^b^	0.66 ^ab^	0.0189
	*C. fumosorosea*	0.66 ^b^	0.70 ^b^	0.58 ^b^	0.92 ^a^	0.0004
	*M. anisopliae*	0.64 ^a^	0.58 ^a^	0.28 ^b^	0.64 ^a^	0.0003

Means with different letters within a row are significantly different (*p* < 0.05).

## Data Availability

Not applicable.

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
