# Peer review of "Laboratory Risk Assessment of Three Entomopathogenic Fungi Used for Pest Control toward Social Bee Pollinators"

_microorganisms, 2022, doi:10.3390/microorganisms10091800_

Round 1

Reviewer 1 Report

The document with comments is attached.

Reviewer 2 Report

This article provides useful information on the risk assessment of the entomopathogenic fungi in social bee pollination both honey bees and stingless bees. The scope and results are very clear. Some minor corrections on pages 5 and 8. 

Reviewer 3 Report

Dear authors,

Thanks for this study. It is crucial to understand the impact of biopesticides on pollinators and other beneficial arthropods. Unfortunately, there are only a few studies on this subject and I think all of them on social pollinators (and solitary wild bees are so important for the pollination of several crops). I think that a sentence on the lack of studies on solitary bees should be included. There are also other studies on the impact of fungi-based biopesticides that should be included in the introduction and / or the discussion sections (some of them very recent and probably not yet available when you prepared the manuscript):

Cappa, F., Baracchi, D., Cervo, R. (2022). Biopesticides and insect pollinators: Detrimental effects, outdated guidelines, and future directions. Science of the Total Environment, 837, 155714

doi: 10.1016/j.scitotenv.2022.155714

Demirozer, O., Uzun, A., Gosterit, A. (2022). Lethal and sublethal effects of different biopesticides on Bombus terrestris (Hymenoptera: Apidae). Apidologie, 53(2): 24. doi: 10.1007/s13592-022-00933-6

Mommaerts, V; Sterk, G; (...); Smagghe, G (2009). A laboratory evaluation to determine the compatibility of microbiological control agents with the pollinator Bombus terrestris. Pest Management Science 65(9): 949-955. doi: 10.1002/ps.1778

Other minor improvements:

Line 139 – Change “methodologies” to “methods”

On Fig. 3 it is difficult to see the different SE bands (change of colour?)

Line 305 – [33] applied B. bassiana topically? Orally?
